# The Study of Structural Features of N- and O-Derivatives of 4,5-Dihydroxyimidazolidine-2-Thione by NMR Spectroscopy and Quantum Chemical Calculations

Liudmila E. Kalichkina [1,†], Alexander V. Fateev [1,2], Polina K. Krivolapenko [1], Kristina A. Isakova [1], Alexey S. Knyazev [1], Victor S. Malkov [1], Abdigali A. Bakibaev [1] and Vera P. Tuguldurova [1,*,†]

1   Faculty of Chemistry, Tomsk State University, 36, Lenin Avenue, Tomsk 634050, Russia
2   Faculty of Chemistry and Biology, Tomsk State Pedagogical University, 60, Kievskaya Street, Tomsk 634061, Russia
*   Correspondence: tuguldurova91@mail.ru
†   These authors contributed equally to this work.

**Abstract:** In the present work, the new N-methylol and O-alkyl derivatives of 4,5-dihydroxyimidazolidine-2-thione (DHIT) are synthesized. The effects of N-alkyl, N-phenyl, N-methylol, and O-alkyl substituents of DHIT on the [13]C and [1]H signals in NMR spectra of the imidazolidine-2-thione ring are systematized using quantum chemical calculations. The shift values of carbon and hydrogen atoms are specific for the geometric isomers of the indicated DHIT derivatives. The chemical shifts of the carbon atoms of the methine groups allows for identifying the cis and trans isomers of the N-alkyl derivatives of DHIT due to the up-field shifts of the cis isomers. The introduction of an alkyl substituent at the N-position of the imidazolidine-2-thione ring leads to the up-field shifts of the carbon atoms of the ring due to the inductive effects of these groups. The ring current effect in the N-phenyl derivatives of DHIT that affects the positions of the carbon signals of the imidazolidine-2-thione ring has been found. The N-methylol derivatives of 4,5-dihydroxyimidazolidine-2-thione have been identified for the first time using 1D and 2D NMR.

**Keywords:** 4,5-dihydroxyimidazolidine-2-thione; N-alkyl derivatives; O-alkyl derivatives; N-methylol derivatives; NMR spectroscopy; quantum chemical calculations

## 1. Introduction

Imidazolidine-2-thiones attract the close attention of researchers due to the established types of biological activity [1–3]. In particular, they exhibit antiproliferative activity against melanoma cells and lung cancer, as well as fungicidal and sedative actions [4]. In addition, imidazolidine-2-thiones are important synthetic precursors of supramolecular structures, i.e., semi-thioglycolurils and semi-thiobambusurils [5,6], which have potential for application in biomedicine, nanoelectronics, and other advanced fields. The presence of sulfur in the structure of these compounds, in contrast to the O-containing counterparts, allows us to use them for the therapy of channelopathies [7–10].

The identification of the compounds containing an imidazolidine-2-thione fragment is mainly carried out by NMR [4–6,10–14]. However, despite the fact that there is a lot of information on the chemical shifts for carbon and hydrogen atoms of imidazolidine-2-thiones in the NMR spectra, a systematic analysis of the effect of substituents on the structural and electronic properties of the imidazolidine-2-thione ring has not been carried out. Thus, our research goal is to specify the general regularities of the effect of substituents in the N and O atoms in DHIT on the C and H chemical shifts of the imidazolidine-2-thione ring in the NMR spectra, as well as to identify the isomeric composition of the final products by comparing the experimental data with the results of quantum chemical calculations.

## 2. Materials and Methods

*2.1. Synthesis*

2.1.1. Synthesis of a Mixture of cis-, trans-4,5-Dihydroxyimidazolidine-2-Thione (**1c**, **1t**)

A total of 1.1 mol of thiourea was dissolved in 1 mol of 40% glyoxal. pH of the resulting solution was adjusted to 5 with 10% $Na_2CO_3$ solution. The mixture was stirred at 50 °C within 30 min and then cooled down to room temperature. The residual was filtrated, and the mixture of cis- and trans-4,5-dihydroxyimidazolidine-2-thione in 1:17 proportion was isolated with a 50% yield. The mixture was recrystallized from water, and the **1t** compound was isolated.

NMR: mixture of **1c**, **1t**: $^{13}C$, δ, ppm: 181.35 (C=S); 79.95 (CH); 182.11 (C=S); 87.22 (CH). $^1H$, δ, ppm: 8.84 (s, NH); 8.66 (s, NH) 4.73 (d, 2H, CH); 5.04 (d, 2H, CH); 6.26 (d, 2H, OH); 5.81 (d, 2H, OH);

**1t**: $^{13}C$, δ, ppm: 182.11 (C=S); 87.22 (CH). $^1H$, δ, ppm: 8.84 (s, NH); 4.71 (d, 2H, CH); 6.26 (d, 2H, OH).

2.1.2. Synthesis of a Mixture of cis- and trans-4,5-Dihydroxy-1,3-Dimethylimidazolidine-2-Thione (**2c**, **2t**)

A total of 1.2 mol of 1,3-dimethylthiourea was dissolved in 1 mol of 40 % glyoxal. pH of the resulting solution was adjusted to 5 with 10 % $Na_2CO_3$ solution. The mixture was stirred at 50 °C within 120 min and then cooled down to room temperature. The residual was filtrated, and the mixture of cis- and trans- isomers in 1:6 proportion was isolated with a 48% yield.

NMR: mixture of **2c**, **2t**: $^{13}C$, δ, ppm: 181.00 (C=S); 87.67 (CH); 181.54 (C=S); 87.93 (CH); 30.80 ($CH_3$); 30.90 ($CH_3$). $^1H$, δ, ppm: 2.98 (s, 6H, $CH_3$); 2.95 (s, 6H, $CH_3$); 4.73 (d, 2H, CH); 4.96 (d, 2H, CH); 6.58 (d, 2H, OH); 6.12 (d, 2H, OH).

2.1.3. Synthesis of a Mixture of cis- and trans-4,5-Dihydroxy-1,3-Diphenylimidazolidine-2-Thione (**4c**, **4t**)

A total of 1 mol of 1,3-diphenylthiourea was mixed with 3 mol of 40% glyoxal solution, and 50 mL of water/IPA (50:50 vol.) was added. pH of the resulting solution was adjusted to 7.5 with 10 % $Na_2CO_3$ solution. The mixture was heated to boiling point, stirred within 60 min, and then cooled down to room temperature. The yellow–brown precipitate was filtrated, and the mixture of cis and trans isomers in 1:11 proportion was isolated with a 52% yield.

NMR: mixture of **4c**, **4t**: $^{13}C$, δ, ppm: 181.07 (C=S); 82.60 (CH); 180.15 (C=S); 89.73 (CH); 126.75, 127.68, 128.47 (Ph), 139.05 (C(Ph)). $^1H$, δ, ppm: 7.31 (t, 2H, Ph);7.44 (t, 4H, Ph), 7.53 (m, 4H, Ph) 5.20 (d, 2H, CH); 5.61 (d, 2H, CH); 7.13 (d, 2H, OH); 6.60 (d, 2H, OH).

2.1.4. Synthesis of a Mixture of cis- and trans-4,5-Dihydroxy-1,3-Bis(Hydroxymethyl)Imidazolidine-2-Thione (**6c**, **6t**)

A total of 1 mol of thiourea was dissolved in 2.5 mol of 40 % glyoxal solution. The mixture was stirred at 40 °C within 120 min. Then, 0.86 mol of 37% formaldehyde solution was added. pH of the resulting solution was adjusted to 8 with 20 % $Na_2CO_3$ solution. The resulting mixture was stirred at 50 °C within 60 min. The green solution was concentrated at 50 °C and 70 mbar, and IPA was added after the concentration stage. The white precipitate was filtrated, and the mixture of cis and trans isomers in 1:16 proportion was isolated with a 51% yield.

NMR: mixture of **6c**, **6t**: $^{13}C$, δ, ppm: 180.43 (C=S); 180.85 (C=S); 85.73 (CH); 78.36 (CH); 66.24 ($CH_2$). $^1H$, δ, ppm: 5.76 (t, 2 OH, $CH_2$-OH); 5.33 (dd, 4H, $CH_2$); 4.63 (dd, 4H, $CH_2$); 5.20 (d, 2H, CH); 5.02 (d, 2H, CH); 6.65 (d, 2H, OH); 6.15 (d, 2H, OH).

2.1.5. Synthesis of a Mixture of cis- and trans-4,5-Dimethoxyimidazolidine-2-Thione (**9c**, **9t**)

A total of 1 mol of trans-4,5-dihydroxyimidazolidine-2-thione, 2 mol of water, and 5 mol of methanol was mixed in the presence of concentrated HCl to adjust pH to 2. The

mixture was stirred at 50 °C within 60 min. After cooling, the precipitate was filtrated, and the mixture of cis and trans isomers in 1:18 proportion was isolated with a 45% yield.

NMR: mixture of **9c**, **9t**: [13]C, $\delta$, ppm: 183.02 (C=S); 184.03 (C=S); 90.93 (CH); 91.78 (CH); 53.53 $CH_3$. [1]H, $\delta$, ppm: 9.45 (s, 2H, NH); 9.27 (s, 2H, NH); 4.68 (d, 2H, CH); 4.83 (d, 2H, CH); 3.24 (s, 6H, $CH_3$)

2.1.6. Synthesis of 4,5-Diethoxyimidazolidine-2-Thione (**10**), 4,5-Dipropoxyimidazolidine-2-Thione (**11**), 4,5-Dibutoxyimidazolidine-2-Thione (**12**)

A total of 1 mol of trans-4,5-dihydroxy imidazolidine-2-thione, 2 mol of water, and 5 mol of alcohol (ethanol, propanol, butanol) was mixed in the presence of hydrochloric acid at pH = 2–4. The mixture was stirred at 50 °C for 1 h. When cooling, a precipitate fell out. The precipitate was filtered and dried in air.

NMR: **10t**: [13]C, $\delta$, ppm: 183.77 (C=S); 90.93 (CH); 15.45 ($CH_3$); 62.59 ($CH_2$). [1]H, $\delta$, ppm: 9.37 (s, 2H, NH); 4.73 (d, 2H, CH); 1.10 (t, 6H, $CH_3$); 3.41(m, 4H, $CH_2$)

NMR: **11t**: [13]C, $\delta$, ppm: 183.68 (C=S); 90.99 (CH); 68.55 ($CH_2$-O); 22.78 ($CH_2$); 10.92 ($CH_3$) [1]H, $\delta$, ppm: 9.34 (s, 2H, NH); 4.73 (d, 2H, CH); 0.84 (t, 6H, $CH_3$); 1.47 (m, 4H, $CH_2$); 3.32 (m, 4H, $CH_2$-O).

NMR: **12t**: [13]C, $\delta$, ppm: 183.66 (C=S); 90.99 (CH); 66.62 ($CH_2$-O); 19.22, 31.58 ($CH_2$); 14.04 ($CH_3$) [1]H, $\delta$, ppm: 9.31 (s, 2H, NH); 4.69 (d, 2H, CH); 0.84 (t, 6H, $CH_3$); 1.47 (m, 8H, $CH_2$); 3.40 (m, 4H, $CH_2$-O).

*2.2. NMR Spectroscopy*

Registration of [1]H, [13]C, DEPT-135, APT, HSQC, HMBC 1H–13C NMR spectra was performed using the Bruker AVANCE III HD 400 MHz NMR spectrometer with the PA BBO 400S1 BBF-H-D-05 Z SP sensor and the BCU temperature control unit, PLC on TTY1 of ELCB 1 autosampler and TopSpin 3.5 pl5 interface. The [1]H and [13]C NMR signals of the samples were assigned to the [1]H and [13]C NMR signals of tetramethylsilane (TMS) (0.0 ppm). The Mestrenova v14.0.0-23239 software was used for spectral analysis.

*2.3. Computational Details*

Geometry optimization of all the structures was carried out using the Gaussian'09 program package [15] installed on the SKIF "Cyberia" supercomputer of Tomsk State University. Hybrid functional M062X [16] and split-valence basis set 6-311+G(d,p) with the addition of diffusion functions for non-hydrogen atoms, a set of d-polarization functions for heavy atoms, and p-functions for hydrogen atoms were used. The geometries of the molecular structures were optimized in the solution using the PCM model. The PCM was applied using a scaled van der Waals surface cavity, with an alpha value of 1.1 and atomic radii modelling using a universal forcefield. DMSO ($\varepsilon$ = 46.826) was a solvent. These settings corresponded to the default parameters of the PCM-type calculations used in Gaussian'09.

Tables S1 and S2 (Supplementary Materials) show the comparison of geometric parameters of optimized **1t** and **4t** molecular structures and experimental data [17,18]. Table S3 (Supplementary Materials) presents the XYZ coordinates for all structures.

Magnetic shielding tensors have been calculated with the gauge including atomic orbitals (GIAO) DFT method as implemented in Gaussian'09. The PBE0/6-311+G(2d,p) level theory was used.

## 3. Results and Discussion

The N- and O-derivatives of 4,5-dihydroxyimidazolidine-2-thione were synthesized (Scheme 1) to study the effect of various substituents in the N and O atoms in the imidazolidine-2-thione ring on chemical shifts in the [13]C and [1]H NMR spectra.

**Scheme 1.** A scheme to synthesize N- and O-derivatives of 4,5-dihydroxyimidazolidine-2-thione.

The N-substituted compounds **1–4** were obtained from the corresponding thioureas and glyoxal through reaction 1 in Scheme 1, as well as through the DHIT reaction with formaldehyde (Scheme 1(3)). The O-substituted compounds were obtained through the 4,5-dihydroxyimidazolidine-2-thione reaction with the corresponding alcohol in an acid medium (Scheme 1(2)). A mixture of cis and trans isomers was formed as a result of the synthesis of N-substituted derivatives **1–4**, while a mixture of geometric isomers was formed only for DHIT dimethyl ether **9** in the case of O-substituted derivatives **9–12**.

Spectra for most of the studied compounds are reported in the literature. However, the spectra for the synthesized compounds were recorded independently in this work with the exception of compound **3**, for which the chemical shifts were taken from the literature [11]. Table 1 lists the chemical shifts for the carbon and hydrogen atoms of the imidazolidine-2-thione ring in cis- and trans-derivatives of DHIT. Further discussion of the results was carried out with respect to the δ values of carbon atoms, since the changes in the δ values of protons in the considered series were insignificant, and appeared only in the case of cis and trans isomers of the same compound. However, it is noteworthy that the proton signals of methine groups for cis isomers were always in a downfield compared to those for trans isomers.

In this work, we consider only the general regularities of changes in the NMR spectra of compounds **1**, **2**, **3**, **4**, **9**, **10**, **11**, and **12**, since the identification of these compounds was described in the literature [17–19]. The exceptions are $^{13}$C, $^{1}$H, DEPT 135, APT, HCQC $^{13}$C-$^{1}$H, HMBC $^{13}$C-$^{1}$H NMR spectra for 4,5-dihydroxy-1,3-bis(hydroxymethyl)imidazolidine-2-thione **6** (synthesis Scheme 1(3)), reported in the present work for the first time.

**Table 1.** Chemical shifts for carbon and hydrogen atoms of the imidazolidine-2-thione ring and energies for cis and trans derivatives of DHIT.

| Compound | Designation | Substitute | | | | $^{13}$C Chemical Shift, ppm | | $^1$H Chemical Shift, ppm | E, a.e. |
|---|---|---|---|---|---|---|---|---|---|
| | | $R_1$ | $R_2$ | $R_3$ | $R_4$ | C=S | CH-CH | CH-CH | |
| 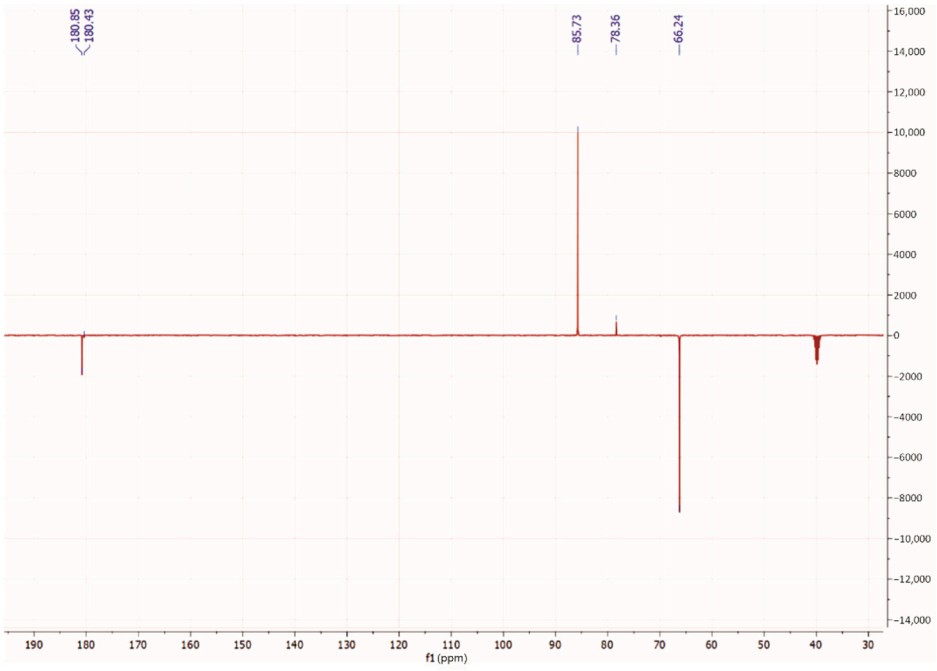 | 1c | H | H | H | H | 180.35 | 79.95 | 5.03 | −776.066928 |
| | 2c | $CH_3$ | $CH_3$ | H | H | 181.00 | 81.67 | 4.96 | −854.664870 |
| | 3c | $C_2H_5$ | $C_2H_5$ | H | H | 179.30 | 79.40 | 5.05 | −933.276457 |
| | 4c | $C_6H_5$ | $C_6H_5$ | H | H | 180.07 | 82.60 | 5.20 | −1238.080635 |
| | 6c | $CH_2OH$ | $CH_2OH$ | H | H | 180.43 | 78.60 | 5.20 | −1005.113298 |
| | 9c | H | H | $CH_3$ | $CH_3$ | 183.02 | 90.93 | 4.68 | −854.634508 |
| | 1t | H | H | H | H | 182.11 | 87.22 | 4.73 | −776.065439 |
| | 2t | $CH_3$ | $CH_3$ | H | H | 181.54 | 87.98 | 4.73 | −854.663088 |
| | 3t | $C_2H_5$ | $C_2H_5$ | H | H | 179.60 | 86.60 | 4.80 | −933.275429 |
| | 4t | $C_6H_5$ | $C_6H_5$ | H | H | 180.15 | 89.73 | 5.61 | −1238.079358 |
| | 6t | $CH_2OH$ | $CH_2OH$ | H | H | 180.85 | 78.35 | 5.05 | −1005.116998 |
| | 9t | H | H | $CH_3$ | $CH_3$ | 184.03 | 91.78 | 4.83 | −854.640392 |
| | 10t | H | H | $C_2H_5$ | $C_2H_5$ | 183.77 | 90.93 | 4.73 | −933.252904 |
| | 11t | H | H | $C_3H_7$ | $C_3H_7$ | 183.68 | 90.99 | 4.73 | −1011.859766 |
| | 12t | H | H | $C_4H_9$ | $C_4H_9$ | 183.66 | 91.03 | 4.68 | −1090.466531 |

### 3.1. Identification of 4,5-Dihydroxy-1,3-Bis(Hydroxymethyl)Imidazolidine-2-Thione (**6**)

The $^{13}$C NMR spectrum for 4,5-dihydroxy-1,3-bis(hydroxymethyl)imidazolidine-2-thione reveals two downfield signals at 180.85 and 180.43 ppm, related to the carbon atom of the thione group; signals at 85.73 and 78.36 ppm, related to the carbon atoms of the ring; and the signal at 66.24 ppm, probably related to the carbon atoms of the methylol substituent. The DEPT 135 and APT (Figure 1) were carried out for unambiguous interpretation of the NMR signals in the $^{13}$C spectrum for 4,5-dihydroxy-1,3-bis(hydroxymethyl)imidazolidine-2-thione.

**Figure 1.** APT spectrum for 4,5-dihydroxy-1,3-bis(hydroxymethyl)imidazolidine-2-thione **6**.

The DEPT 135 data allow us to identify different types of substitution of carbon atoms. We observe that that the signals at 180.85 and 180.43 ppm are related to the thione group. It is found that the signal at 66.24 ppm directed in the same direction with the thione group corresponds to the methylene group, and using APT, the 85.73 and 78.36 ppm signals directed in the opposite direction are attributed to the methine carbon atoms (Figure 1).

Figure 2 shows the proton spectrum for compound **6**.

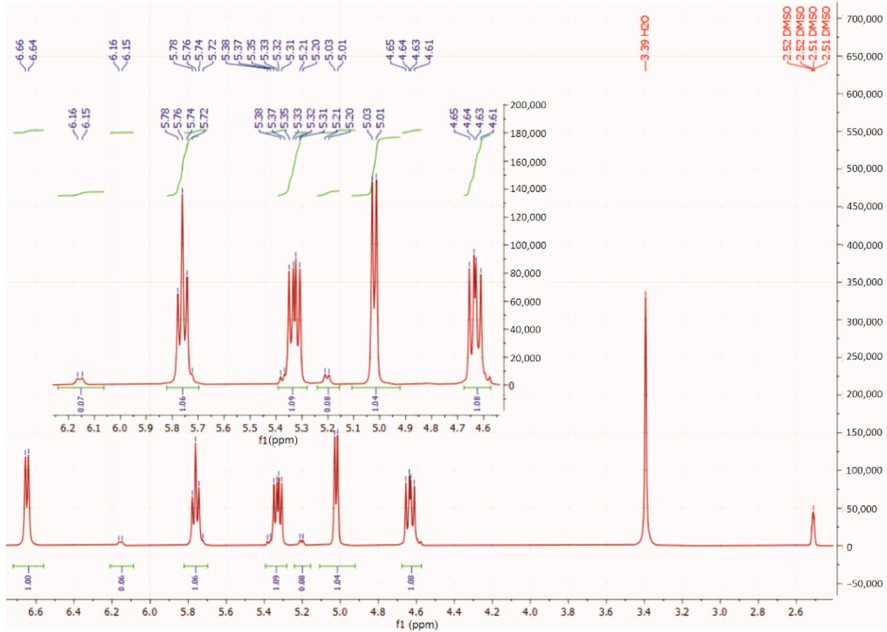

**Figure 2.** $^1$H spectrum for 4,5-dihydroxy-1,3-bis(hydroxymethyl)imidazolidine-2-thione **6**.

The spectrum features the signals of various shapes. The doublets correspond to protons of hydroxyl groups bonded with the imidazolidine-2-thione ring, as well as the protons of the methine group based on the structure of compound **6**. Thus, the doublet downfield signal with δ 6.65 ppm most likely is referred to the hydroxyl group and the doublet at 5.02 ppm corresponds to the protons of the methine groups. The ratio of the integral intensities is 1:1. Two signals in the form of a doublet of doublets at 5.38–5.20 ppm and 4.65–4.62 ppm correspond to methylene groups with a ratio of integrated intensities of 1:1.

The triplet signal at 5.78–5.72 ppm can be assigned to the protons of the hydroxyl group of the methylol substituents. The presence of two sets of $^1$H signals with different integral intensities indicates the possible formation of geometric isomers of compound **6**. Therefore, the downfield doublet at 6.14 ppm corresponds to the protons of the hydroxyl group of one of the geometric isomers of compound **6**, and the doublet at 5.20 ppm corresponds to hydrogen atoms of the methine group of the same isomer with an integral ratio of 1:1.

Two-dimensional spectra for $^{13}$C–$^1$H HMQC and $^{13}$C–$^1$H HMBC were recorded to determine the correlations between H and C atoms (Figures 3–5).

The $^{13}$C–$^1$H HMQC NMR spectrum shows direct correlations between C and H atoms. The $^{13}$C–$^1$H HMBC spectrum that characterizes long-range interactions of carbon atoms with protons allows for identifying the bonding order of carbon atoms in a molecule. The signals of methine groups have a direct correlation: $^{13}$C-$^1$H: 86.26–5.02 ppm and 78.36–5.20 ppm (Figure 3). The shielding constant of the cis isomer is less than that of the trans isomer. Thus, the signal at 5.02 ppm (J = 6.1 Hz) can be attributed to the protons of the methine group of the trans isomer **6t**, while the signal at 5.20 ppm (J = 6.0 Hz) can be assigned to the cis isomer **6c**. The signals at 5.02 ppm for methine group protons of compound **6t** have a long-range correlation with the carbon atom signals at 180.85 ppm, 66.2 ppm, and 86.26 ppm (Figures 4 and 5). Two signals in the form of a doublet of doublets at 5.38–5.20 ppm and 4.65–4.62 ppm have a direct correlation with the signal of the carbon

atom at 66.81 ppm and can be assigned to the protons of the methylene group (Figure 3). The proton signals of the methylene group at 5.38–5.20 ppm and 4.65–4.62 ppm have a long-range correlation with the carbon signals at 180.85 and 86.65 ppm of thione and methine groups, respectively, in compound **6t** (Figures 4 and 5).

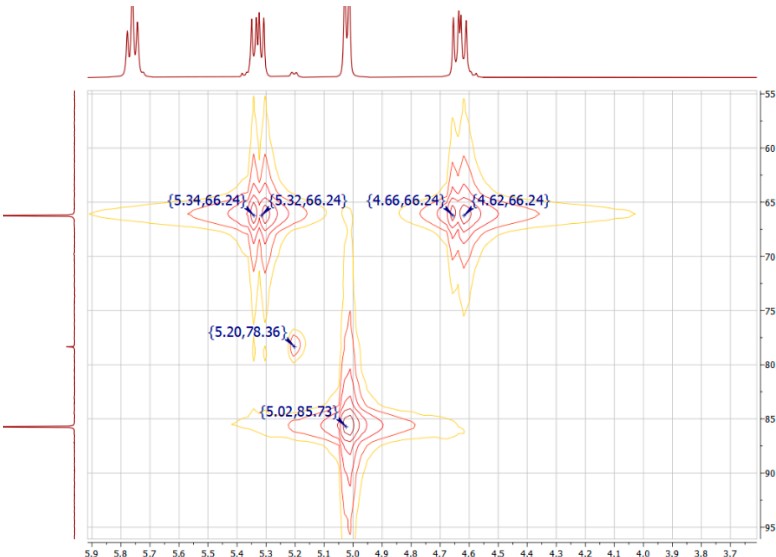

**Figure 3.** $^{13}$C—$^1$H HMQC spectrum for 4,5-dihydroxy-1,3-bis(hydroxymethyl)imidazolidine-2-thione **6**.

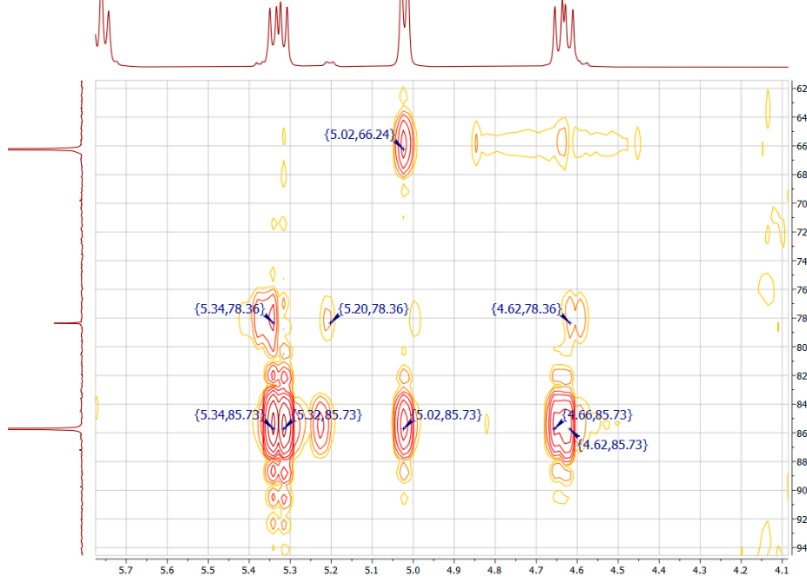

**Figure 4.** The range 90–60 ppm—5.7–4.1 ppm of $^{13}$C—$^1$H HMBC spectrum for 4,5-dihydroxy-1,3-bis(hydroxymethyl)imidazolidine-2-thione **6**.

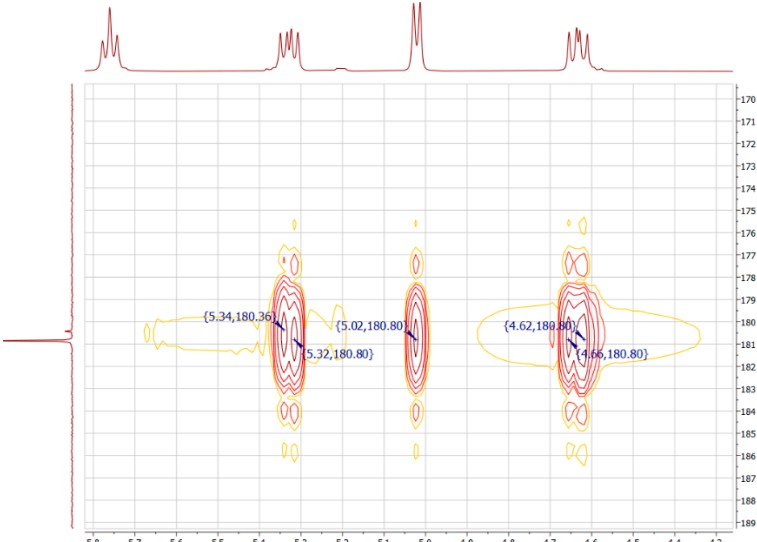

**Figure 5.** The range 190–170 ppm—5.8–4.4 ppm of $^{13}$C—$^1$H HMBC spectrum for 4,5-dihydroxy-1,3-bis(hydroxymethyl)imidazolidine-2-thione **6**.

Thus, all signals in the NMR obtained spectra were identified. A mixture of geometric isomers with a ratio of cis-**6c** to trans-**6t** of 1:16 according to the experimental proton spectrum (Figure 2) is formed as a result of compound **6** synthesis. The prevalence of the trans isomer **6t** is confirmed by the calculated data on the energy of the molecules, since the cis isomer **6c** lies above **6t** by 2.3 kcal/mol (Table 1).

The compound **6t** can form intermolecular and intramolecular hydrogen bonds in the solution due to the presence of four hydroxyl and thione groups in the molecule (Figure 6).

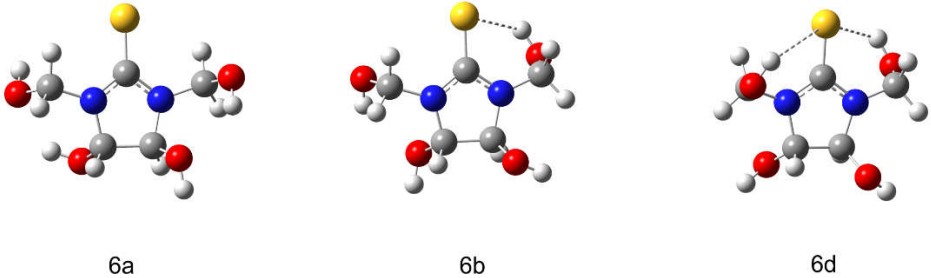

**Figure 6.** Formation of intramolecular hydrogen bonds in 4,5-dihydroxy-1,3-bis(hydroxymethyl)imidazolidine-2-thione **6t**.

The calculated $^{13}$C NMR spectra for compound **6t** for the structures in the series **6a**, **6b**, and **6d** show an up-field move of the carbon signal in the thione group (184.93 ppm, 184.36 ppm, 181.43 ppm) due to the formation of intramolecular hydrogen bonds as shown in Ref. [20]. This fact approximates the calculated values to the experimental value to 180.85 ppm, which reflects the presence of a network of hydrogen bonds in compound **6t** in a real solution.

*3.2. The N-Derivatives of DHIT: General Regularities in the NMR Spectra*
3.2.1. Geometric Isomerism

It is known that a mixture of geometric isomers is formed in reactions of thiourea and its N-alkyl derivatives with glyoxal (Scheme 1(1)). The ratio of isomers varies depending on the experimental conditions with a prevalence of trans isomers [12–14]. At the same time, the quantum chemical calculations of molecules **1**, **2**, and **3** show that the energies of cis isomers (**1c**, **2c**, **3c**) are lower than those of trans isomers (**1t**, **2t**, **3t**) with differences of 0.9; 1.1; and 0.6 kcal/mol, respectively (Table 1). The incompatibility between this experimental

fact and the calculated data can be explained by considering the structure of the geometric isomers of DHIT (Figure 7).

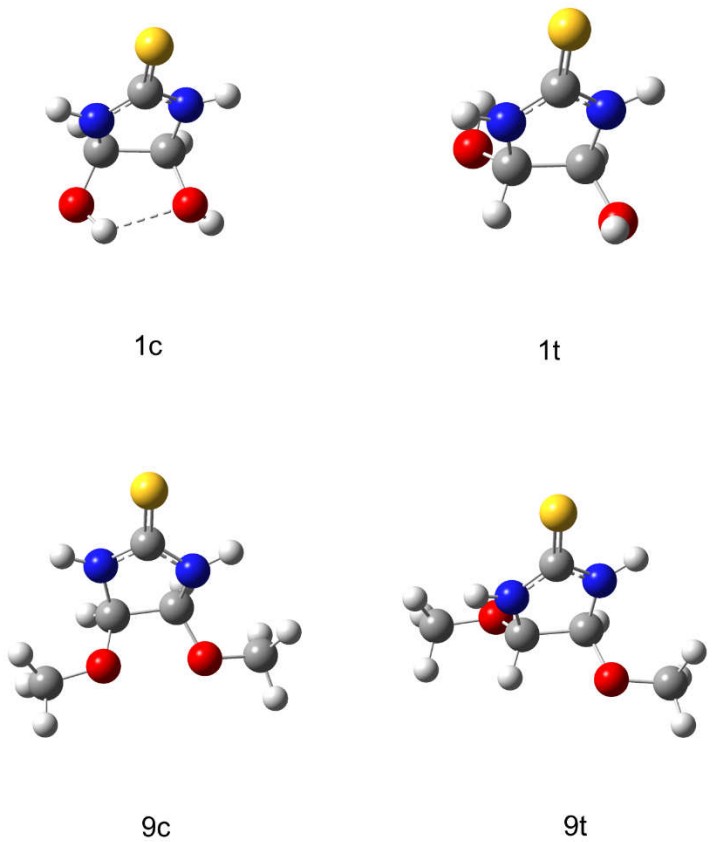

**Figure 7.** The cis and trans structures of DHIT **1** and its dimethyl ether **9**.

Figure 7 shows that a hydrogen bond is formed between the hydroxyl groups in the cis position relative to the imidazolidine-2-thione ring that stabilizes the molecule and leads to a general decrease in its energy. Additional evidence for this interpretation is that the energy of the cis-dimethyl ether DHIT **9c** is by 3.7 kcal/mol higher than that of trans isomer **9t** (Table 1), because replacing hydroxyl groups with methoxy ones breaks hydrogen bonds. The trans isomer exists in excess in the final products of the experimental synthesis.

The NMR allows reliable identification of the geometric isomers of DHIT and δ $^{13}$C for cis DHIT downfield compared to trans DHIT (Table 1). The Mulliken charges (e) on the carbon atoms of the imidazolidine-2-thione ring were calculated using quantum chemical approach to explain this experimental fact (Figure 8).

The charges on 3C and 4C atoms of compound **1t** have the same value and are equal to −0.141 due to the location of the hydroxyl groups on the opposite sides of the imidazolidine-2-thione ring, resulting in delocalization of electron density on the methine carbon atoms 3C and 4C. For cis DHIT **1c**, the electron density is localized on the 3C atom and has the smallest charge of −0.258. The charge on other 4C carbon atom is 0.029. Thus, the higher localization of the electron density on one of the carbon atoms and the abovementioned intramolecular hydrogen bonding between the cis hydroxyl groups causes the imidazolidine-2-thione ring to exist in some steric "tension". This effect explains the shielding of the methine carbon atoms of the ring and their up-field value of chemical shift at 79.95 ppm for the cis isomer, as well as the de-shielding of the same atoms in the trans DHIT **1t**, and, as a result, a downfield of the chemical shift at 87.22 ppm. The observed effect is of a general nature for all geometric isomers of DHIT derivatives indicated in the present work.

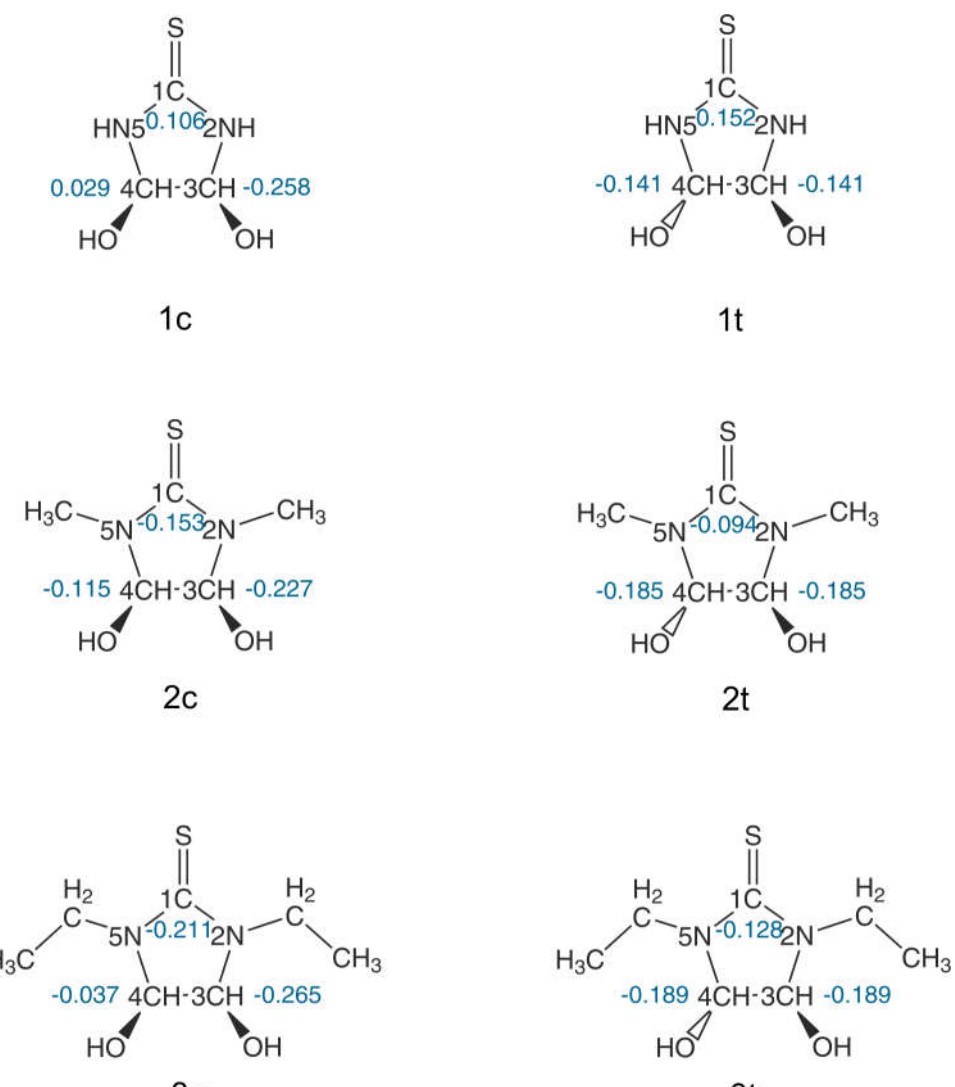

**Figure 8.** The calculated Mulliken charges (e) on the atoms of compounds **1–3**.

### 3.2.2. Nature of the Substituents

The DHIT molecule **1** can be depicted as resonant structures a, b, and c (Scheme 2). The electron density is delocalized between atoms 1C, 5N, and 2N. The distribution of charges in the thioamide fragment in compounds **1c** and **1t** is associated with the 2p-3p orbital interaction between the 1C and S atoms. Since the size of the 3p orbital of the S atom is larger than that of the carbon 2p orbital, which reduces the efficiency of overlapping between them, this contributes to the concentration of a positive charge on 1C and a negative charge on S [21].

**Scheme 2.** The DHIT resonance structures.

The distribution of charges in the thioamide fragment in compounds **1c** and **1t** is associated with the 2p-3p orbital interaction between the 1C and S atoms. Since the size of the 3p orbital of the S atom is larger than the one of carbon 2p orbital, which reduces the efficiency of overlapping between them, and this contributes to the concentration of a positive charge on 1C and a negative charge on S [21]. Therefore, when replacing the hydrogen atom in the 2N and 5N nitrogen atoms with alkyl substituents, i.e., methyl, ethyl, we experimentally observe that the shielding of the thione group increases with an increase in the alkyl radical, and δ values decrease in comparison with those for unsubstituted DHIT (Table 1).

The δ value is 181.35 ppm, 181.00 ppm, and 179.30 ppm in the series for cis DHITs **1c**, **2c**, and **3c**, respectively; for trans isomers **1t**, **2t**, and **3t** the δ equals to 182.11 ppm, 181.54 ppm, and 179.60 ppm, respectively (Table 1). Obviously, the observed changes in chemical shifts are associated with an increase in the electron density on the 1C atom due to the positive inductive effect of alkyl groups. This experimental fact was confirmed by quantum chemical calculations of the charges on the 1C atom in the DHIT molecule (Figure 8), and an anti-batic charge decrease occurs with an increase in the alkyl radical. In the series of compounds **1c**, **2c**, and **3c**, the values of the charges on the 1C atom are 0.106; −0.153; and −0.211, respectively, and for trans isomers in the series **1t**, **2t**, and **3t,** the values on 1C correspond to 0.152; −0.094; and −0.128, respectively.

An identical dependence is observed when considering the effect of N-alkyl substitution on the $^{13}$C NMR of methine carbon atoms 4C and 3C in the series for cis and trans isomers of compounds **1**, **2**, and **3**. The δ value for the methine carbon atoms of the ring first slightly increases with the introduction of methyl substituents, then decreases with the introduction of ethyl substituents (Table 1). The chemical shifts are equal to 79.95 ppm, 81.67 ppm, and 79.40 ppm in the series of cis isomers, and 87.22 ppm, 87.93 ppm, and 86.60 ppm for trans isomers. This change in the character of δ is probably due to the smaller inductive effect of the methyl group compared to the ethyl group. The methyl value is not enough to shield the methine carbon atoms. However, when ethyl substituents with higher inductive effect are introduced, the shielding of the methine carbon atoms is experimentally observed. The proposed interpretation of the difference in chemical shifts of CH groups is consistent with the calculated values of the charges on methine carbon atoms with a minimum value in molecules **3c** and **3t** (Figure 8).

Conjugation of the thione group with phenyl substituents leads to additional shielding and a decrease in its δ value when replacing alkyl substituents with phenyl ones. However, methine carbon atoms are deshielded in this case, and their δ is downfield. The experimentally established fact is clearly illustrated in Figure 9.

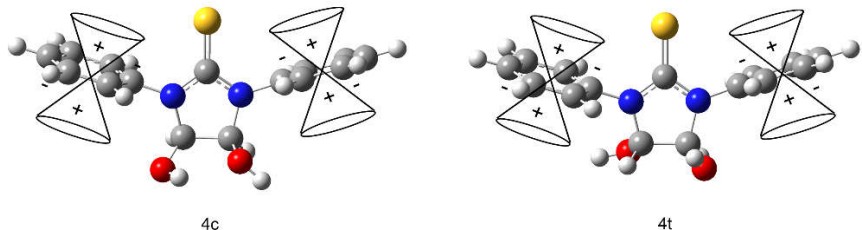

4c                                                                      4t

**Figure 9.** Location of anisotropy cone of phenyl rings in structures of cis-**4c** and trans-**4t** 4,5-dihydroxy-1,3-diphenylimidazolidine-2-thione.

Figure 9 shows that the phenyl rings are located relative to the imidazolidine-2-thione ring in such a way that the anisotropy cones they create shield the thione group, while the methine groups are in the de-shielding region (Table 1).

### 3.3. The O-Derivatives of DHIT: General Regularities in the NMR Spectra

Nature of Substituents

The compounds **9–12** are products of O-substitution of a hydrogen atom by alkyl groups (from methyl to butyl). A mixture of cis- and trans-alkoxy derivatives of DHIT is formed only in the case of dimethyl ether of DHIT (structures **9t** and **9c**). Most likely, this is due to steric hindrances as a result of an increase in the size of the substituent, as well as the lack of stabilization of the cis position by hydrogen bonds described above. The 3C and 4C carbons of the methine groups, as well as the thione group in compounds **9t–12t**, are deshielded with the introduction of alkyl substituents in oxygen atoms in comparison with the unsubstituted DHIT **1t** as shown experimentally (Table 1). In this case, the calculated charges on carbon atoms do not give satisfactory regularities, which indicates the influence of other factors (solvent effect, absence of hydrogen bonds, etc.).

### 4. Conclusions

1. For the first time, the NMR spectra for 4,5-dihydroxy-1,3-bis(hydroxymethyl) imidazolidine-2-thione were reported and discussed. This allowed for identifying cis and trans isomers in a 1:16 proportion. The formation of intramolecular hydrogen bonds between methylol OH groups and the C=S group in 4,5-dihydroxy-1,3-bis(hydroxymethyl) imidazolidine-2-thione was established;

2. Quantum chemical calculations show that N-alkylated cis isomers of DHIT have lower molecular energy in comparison with N-alkylated trans isomers because of intramolecular hydrogen bonding between two OH groups;

3. It is found that chemical shifts of carbon atoms in methine groups make it possible to identify the cis and trans isomers of N-alkylated derivatives of DHIT due to the up-field shift of the cis isomer signals associated with a special electron density redistribution for these structures;

4. N-alkylation of imidazolidine-2-thione ring leads to an up-field shift of carbon atoms of thione and methine groups signals in $^{13}$C NMR spectra, which is due to the induction effects of these groups and is confirmed with quantum chemical calculations of the Mulliken charges on the ring atoms;

5. The conjugation of the thione group with N-phenyl substituents of DHIT leads to additional shielding and a decrease in the chemical shift of carbon atoms, but methine carbon atoms are deshielded in this case and their signals shift in a weak-field due to benzene ring current;

6. O-alkylation of DHIT leads to de-shielding of carbon atoms of thione and methine groups in comparison with the unsubstituted DHIT. This regularity cannot be explained with quantum chemical calculations.

**Supplementary Materials:** The following supporting information can be downloaded at: https://www.mdpi.com/article/10.3390/magnetochemistry9010015/s1, Table S1: The comparison of geometric parameters of 1t optimized molecular structure and experimental data [17]; Table S2: The comparison of geometric parameters of 4t optimized molecular structure and experimental data [18]; Table S3: Cartesian coordinates of the optimized structures.

**Author Contributions:** Conceptualization, L.E.K. and V.P.T.; methodology, A.A.B., V.S.M., and A.S.K.; software, V.P.T., P.K.K., K.A.I., and A.V.F.; validation, L.E.K., P.K.K., and K.A.I.; formal analysis, V.P.T., L.E.K., A.A.B., and A.V.F.; investigation, L.E.K. and V.P.T.; resources, A.S.K. and V.S.M.; data curation, V.P.T., L.E.K., A.A.B., and A.V.F.; writing—original draft preparation, L.E.K. and V.P.T.; writing—review and editing, A.S.K., A.A.B., and V.S.M.; visualization, P.K.K. and K.A.I.; supervision, A.A.B. and A.S.K.; project administration, L.E.K. and V.P.T.; funding acquisition, V.P.T., L.E.K., A.A.B., and A.V.F. All authors have read and agreed to the published version of the manuscript.

**Funding:** This study was supported by the Tomsk State University Development Programme (Priority 2030).

**Institutional Review Board Statement:** Not applicable.

**Informed Consent Statement:** Not applicable.

**Data Availability Statement:** The data that support the findings of this study are available within the article and the Supplementary Materials. Further data are available from the corresponding author upon reasonable request.

**Acknowledgments:** The authors thank Mikhail Salaev (Tomsk State University) for the language review. The authors also thank Oleg Kotelnikov (Tomsk State University) for recording NMR spectra.

**Conflicts of Interest:** The authors declare no conflict of interest.

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
