# Peer review of "The Study of Structural Features of N- and O-Derivatives of 4,5-Dihydroxyimidazolidine-2-Thione by NMR Spectroscopy and Quantum Chemical Calculations"

_magnetochemistry, doi:10.3390/magnetochemistry9010015_

Round 1
Reviewer 1 Report
In this paper, the new N-methylol and O-alkyl derivatives of 4,5-dihydroxyimidaz- 11 olidine-2-thione (DHIT) are reported. The effect of N-alkyl, N-phenyl, N-methylol, and O-alkyl 12 substituents of DHIT on the 13C, 1H signals in NMR spectra of the imidazolidine-2-thione ring is systematized using quantum-chemical calculations performed at the B3LYP level with the split-valence basis set 6-311+G(d,p). I do not understand the phrase "The shielding constants were calculated by th GIAO method using the 6-311+G(2d,p) basis set since the signal of tetramethylsilane calculated in such a basis set is taken as a reference in the Gaussian’09 program". What authors do mean under the "The shielding constants were calculated by the GIAO method"? What particular functional was used in those calculations? I strongly recommend to reoptimise geometries with using the Minnesota M06-2X functional which was specifically introduced for geometry optimisations. At that, NMR calculations should be performed with using the PBE0 functional, which is highly recommended for the calculations of NMR shielding constants (chemical shifts). The rest of the submission sounds reasonable.
Author Response
Dear Reviewer,
We are thankful for your attention to this manuscript. This careful review helped us to better understand the results obtained. All comments and recommendations helped us to improve this manuscript. The comments and corrections are presented below.
Reviewer 1
Reviewer comment
In this paper, the new N-methylol and O-alkyl derivatives of 4,5-dihydroxyimidaz- 11 olidine-2-thione (DHIT) are reported. The effect of N-alkyl, N-phenyl, N-methylol, and O-alkyl 12 substituents of DHIT on the 13C, 1H signals in NMR spectra of the imidazolidine-2-thione ring is systematized using quantum-chemical calculations performed at the B3LYP level with the split-valence basis set 6-311+G(d,p).
I do not understand the phrase "The shielding constants were calculated by th GIAO method using the 6-311+G(2d,p) basis set since the signal of tetramethylsilane calculated in such a basis set is taken as a reference in the Gaussian’09 program". What authors do mean under the "The shielding constants were calculated by the GIAO method"? What particular functional was used in those calculations
Author response
Probably, we used an unfortunate wording to describe the NMR calculation method in Section 2.3. Initially, NMR calculations were carried out at the B3LYP/6-311+G(2d,p) level of theory with the GIAO functions. According to your recommendations, we used the PBE0 functional. Thus, in the manuscript the phrase “To calculate the NMR spectra, the structure geometry optimized at the B3LYP/6-311+G(d,p) level theory was used. The shielding constants were calculated by the GIAO method using the 6-311+G(2d,p) basis set since the signal of tetramethylsilane calculated in such a basis set is taken as a reference in the Gaussian’09 program" was replaced with "Magnetic shielding tensors have been calculated with the GIAO (gauge-including atomic orbitals) DFT method as implemented in the Gaussian09. The PBE0/6-311+G(2d,p) level theory was used"
Reviewer comment
I strongly recommend to reoptimise geometries with using the Minnesota M06-2X functional which was specifically introduced for geometry optimisations. At that, NMR calculations should be performed with using the PBE0 functional, which is highly recommended for the calculations of NMR shielding constants (chemical shifts). The rest of the submission sounds reasonable.
Author response
We agree with the reviewer. The geometries of all structures were reoptimized at the m062x/6-311+G(d,p) level of theory and NMR calculations were performed at the PBE0/6-311+G(2d,p).
The consistency of geometric parameters of optimized molecular structures 1t and 4t and experimental data given in the revised Supplementary Materials has indeed improved slightly. A comparison of the geometric parameters obtained by different methods is presented below.
1t
|
Parameter |
Experimental data |
Calculation data B3LYP/6-311+G(d,p) |
Calculation data m062x /6-311+G(d,p) |
|
|
|||
|
Bond, Ǻ |
|
||
|
1S-3C |
1.6909 |
1.684 |
1.674 |
|
2N-3C |
1.3421 |
1.356 |
1.355 |
|
2N-1C |
1.4523 |
1.450 |
1.449 |
|
4N-3C |
1.3453 |
1.356 |
1.355 |
|
4N-5C |
1.4559 |
1.450 |
1.450 |
|
2O-1C |
1.4045 |
1.411 |
1.400 |
|
3O-5C |
1.4087 |
1.411 |
1.400 |
|
5C-1C |
1.5440 |
1.564 |
1.548 |
|
Angle, ° |
|
||
|
2N3C4N |
109.0 |
107.36 |
107.23 |
|
2O1C5C3O |
139.6 |
126.65 |
141.65 |
4t
|
Parameter |
Experimental data |
Calculation data B3LYP/6-311+G(d,p) |
Calculation data m062x /6-311+G(d,p) |
|
|
|
||||
|
Bond, Ǻ |
|
|||
|
S1—C1 |
1.669 |
1.682 |
1.672 |
|
|
O1—C2 |
1.427 |
1.413 |
1.401 |
|
|
O2—C3 |
1.416 |
1.409 |
1.397 |
|
|
N1—C1 |
1.357 |
1.366 |
1.365 |
|
|
N1—C4 |
1.444 |
1.432 |
1.428 |
|
|
N1—C2 |
1.459 |
1.462 |
1.460 |
|
|
N2—C1 |
1.373 |
1.367 |
1.360 |
|
|
N2—C10 |
1.433 |
1.433 |
1.427 |
|
|
N2—C3 |
1.454 |
1.462 |
1.457 |
|
|
C2—C3 |
1.526 |
1.544 |
1.534 |
|
|
C4—C9 |
1.374 |
1.396 |
1.391 |
|
|
C4—C5 |
1.396 |
1.396 |
1.393 |
|
|
C5—C6 |
1.394 |
1.394 |
1.390 |
|
|
C6—C7 |
1.368 |
1.395 |
1.393 |
|
|
C7—C8 |
1.387 |
1.395 |
1.392 |
|
|
C8—C9 |
1.394 |
1.394 |
1.392 |
|
|
C10—C15 |
1.390 |
1.395 |
1.392 |
|
|
C10—C11 |
1.393 |
1.395 |
1.391 |
|
|
C11—C12 |
1.389 |
1.394 |
1.392 |
|
|
C12—C13 |
1.386 |
1.395 |
1.392 |
|
|
C13—C14 |
1.384 |
1.395 |
1.393 |
|
|
C14—C15 |
1.384 |
1.394 |
1.390 |
|
|
Angle, ° |
|
|||
|
C1—N1—C4 |
126.4 |
125.83 |
125.31 |
|
|
C1—N1—C2 |
110.97 |
112.22 |
111.24 |
|
|
C4—N1—C2 |
122.5 |
121.85 |
120.15 |
|
|
C1—N2—C10 |
126.6 |
125.63 |
125.85 |
|
|
C1—N2—C3 |
111.1 |
112.49 |
112.11 |
|
|
C10—N2—C3 |
119.53 |
121.86 |
121.26 |
|
|
N1—C1—N2 |
108.0 |
107.99 |
107.57 |
|
|
N1—C1—S1 |
125.46 |
126.00 |
126.09 |
|
|
N2—C1—S1 |
126.6 |
126.00 |
126.33 |
|
|
O1—C2—N1 |
111.04 |
113.07 |
112.31 |
|
|
O1—C2—C3 |
110.76 |
108.40 |
108.15 |
|
|
N1—C2—C3 |
102.8 |
102.78 |
101.91 |
|
|
O2—C3—N2 |
112.5 |
113.16 |
112.44 |
|
|
O2—C3—C2 |
113.8 |
114.38 |
114.15 |
|
|
N2—C3—C2 |
101.37 |
102.60 |
101.64 |
|
|
C9—C4—C5 |
119.9 |
120.41 |
120.82 |
|
|
C9—C4—N1 |
119.9 |
120.16 |
119.00 |
|
|
C5—C4—N1 |
120.1 |
119.39 |
120.13 |
|
|
C6—C5—C4 |
119.6 |
119.72 |
119.42 |
|
|
C7—C6—C5 |
120.9 |
120.08 |
120.10 |
|
|
C6—C7—C8 |
119.0 |
120.01 |
120.14 |
|
|
C7—C8—C9 |
121.1 |
120.16 |
120.06 |
|
|
C4—C9—C8 |
119.5 |
119.62 |
119.47 |
|
|
C15—C10—C11 |
120.3 |
120.42 |
120.83 |
|
|
C15—C10—N2 |
120.0 |
119.38 |
120.30 |
|
|
C11—C10—N2 |
119.6 |
120.17 |
118.85 |
|
|
C12—C11—C10 |
119.4 |
119.65 |
119.51 |
|
|
C13—C12—C11 |
119.9 |
120.15 |
120.03 |
|
|
C14—C13—C12 |
120.7 |
119.99 |
120.09 |
|
|
C15—C14—C13 |
119.6 |
120.08 |
120.16 |
|
|
C14—C15—C10 |
120.1 |
119.72 |
119.39 |
|
The calculated NMR chemical shifts are discussed only in section 3.1 on page 9 for structures 6a, 6b, 6d. The obtained values (184.93, 184.36, 181.43 ppm) are much closer to the experimental values (180.85 ppm) and the upfield move of the signals with the formation of hydrogen bonds are preserved.
The corresponding changes were made throughout the text of the revised manuscript and the Supplementary Materials, and did not affect the presentation of the results and conclusions made.
Sincerely yours and on behalf of the authors,
Dr. Vera P. Tuguldurova
e-mail: tuguldurova91@mail.ru

Reviewer 2 Report
The study of N and O- derivatives of 4,5-dihydroxyimidazolidine-2-thione using spectroscopic and theoretical information is carried out. The results are exciting and nicely presented. Therefore, I recommend the publication of the present manuscript.
However, there is an issue that I would like to point out. On page 8, the authors claim that they observed a 1:16 ratio of cis-6c to trans-6t. This implies a difference of about 1.6 kcal/mol between both conformers. Their calculated value is 2.2, which is pretty close, considering the calculation method they used (DFT and PCM with DMSO as a solvent). However, 1c, 2c, and 3c are lower in energy than 1t, 2t, and 3t isomers, while experiments show that trans is prevalent. My first suggestion is: Can the authors provide an experimental trans-to-cis ratio? In any case, the energy differences are very small (0.7, 1.1, and 0.2), so maybe these discrepancies are due to the quantum mechanical methods used. So, my second suggestion is to analyze whether a different DFT method or a different choice of solvent, or inclusion of thermal effects on the calculation of the free energies can alter these results. The same applies to the calculated charges on carbon atoms described on page 12.
Author Response
Dear Reviewer,
We are thankful for your attention to this manuscript. This careful review helped us to better understand the results obtained. All comments and recommendations helped us to improve this manuscript. The comments and corrections to the reviews are presented below.
Reviewer 2
Reviewer comment
The study of N and O- derivatives of 4,5-dihydroxyimidazolidine-2-thione using spectroscopic and theoretical information is carried out. The results are exciting and nicely presented. Therefore, I recommend the publication of the present manuscript.
However, there is an issue that I would like to point out. On page 8, the authors claim that they observed a 1:16 ratio of cis-6c to trans-6t. This implies a difference of about 1.6 kcal/mol between both conformers. Their calculated value is 2.2, which is pretty close, considering the calculation method they used (DFT and PCM with DMSO as a solvent). However, 1c, 2c, and 3c are lower in energy than 1t, 2t, and 3t isomers, while experiments show that trans is prevalent. My first suggestion is: Can the authors provide an experimental trans-to-cis ratio? In any case, the energy differences are very small (0.7, 1.1, and 0.2), so maybe these discrepancies are due to the quantum mechanical methods used.
Author response
Thank you for your high opinion about our work and for your comments.
The ratio of cis-6c to trans-6t equal to 1:16 was obtained by the assignment of the integral intensities of the signals of the methine protons of the ring in the experimental spectrum, i.e., this trans-to-cis ratio is experimental. We have added the word "experimental" to the second paragraph of page 8 for clarity.
Modeling the structures 1c, 2c, 3c shows that they have lower energies due to the formation of hydrogen bonds between two neighboring hydroxyl groups of the ring as shown in Figure 7. The absence of such a bond, e.g., in structure 9, leads to an increase in the energy of cis-conformer relative to the trans one. Modeling of these structures does not take into account the presence of a large number of hydrogen bonds in a real solution.
The structure 6t even under our simulation conditions is more favorable, because it contains 2 intramolecular hydrogen bonds between the hydroxyl substituents of the ring and the OH groups of the methylol substituents. See pictures below.
|
6c |
6t |
Reviewer comment
So, my second suggestion is to analyze whether a different DFT method or a different choice of solvent, or inclusion of thermal effects on the calculation of the free energies can alter these results. The same applies to the calculated charges on carbon atoms described on page 12.
Author response
We used other method to optimize the geometry of the structures (m062x/6-311+g(d,p)), and also, based on your recommendation, included various corrections in the energy analysis. The results are shown in the table below.
|
Designation |
Substitute |
b3lyp/6-311+g(d,p) |
m062x/6-311+g(d,p) |
|||||||||
|
R1 |
R2 |
R3 |
R4 |
E, a.e. |
ΔE, kcal/mol |
E, a.e |
ΔE, kcal/mol |
G, a.e. |
ΔG, kcal/mol |
H, a.e. |
ΔH, kcal/ |
|
|
1c |
H |
H |
H |
H |
-776.24466 |
|
-776.066928 |
|
-775.991756 |
|
-775.950620 |
|
|
2c |
CH3 |
CH3 |
H |
H |
-854.87962 |
|
-854.664870 |
|
-854.537558 |
|
-854.489353 |
|
|
3c |
C2H5 |
C2H5 |
H |
H |
-933.53174 |
|
-933.276457 |
|
-933.096732 |
|
-933.041361 |
|
|
6c |
СH2OH |
СH2OH |
H |
H |
-1005.3717 |
|
-1005.113298 |
|
-1004.976727 |
|
-1004.924509 |
|
|
9c |
H |
H |
CH3 |
CH3 |
-854.85431 |
|
-854.634508 |
|
-854.507137 |
|
-854.458765 |
|
|
1t |
H |
H |
H |
H |
-776.24348 |
0.7 |
-776.065439 |
0.9 |
-775.990594 |
0.7 |
-775.949220 |
0.9 |
|
2t |
CH3 |
CH3 |
H |
H |
-854.87788 |
1.1 |
-854.663088 |
1.1 |
-854.536865 |
0.4 |
-854.487595 |
1.1 |
|
3t |
C2H5 |
C2H5 |
H |
H |
-933.53145 |
0.2 |
-933.275429 |
0.6 |
-933.096295 |
0.3 |
-933.040509 |
0.5 |
|
6t |
СH2OH |
СH2OH |
H |
H |
-1005.3751 |
-2.2 |
-1005.116998 |
-2.3 |
-1004.980000 |
-2.1 |
-1004.928181 |
-2.3 |
|
9t |
H |
H |
CH3 |
CH3 |
-854.86032 |
-3.8 |
-854.640392 |
-3.7 |
-854.513551 |
-4.0 |
-854.464727 |
-3.7 |
As can be seen, the use of another functional as well as the inclusion of corrections for enthalpy and Gibbs energy does not fundamentally affect the difference in the energies of cis- and trans- conformers. The electronic energies obtained at the level of theory m062x/6-311+g(d,p) were used for discussion in the revised manuscript.
The values of the charges on carbon atoms, presented in Figure 8, have changed, but the features discovered earlier have been preserved.
The corresponding changes were made throughout the text of the revised manuscript and the Supplementary Materials did not affect the presentation of the results and the conclusions.
Sincerely yours and on behalf of the authors,
Dr. Vera P. Tuguldurova
e-mail: tuguldurova91@mail.ru

Round 2
Reviewer 1 Report
All my suggestions were treated adequately.